# Association between pertussis vaccination in infancy and childhood asthma: A population-based record linkage cohort study

Gladymar Pérez Chacón[1,2], Parveen Fathima[1,3], Mark Jones[3], Marie J. Estcourt[3], Heather F. Gidding[4,5,6,7], Hannah C. Moore[1,2], Peter C. Richmond[1,8], Tom Snelling[1,2,3]*

1 Wesfarmers Centre of Vaccines and Infectious Diseases, Telethon Kids Institute, University of Western Australia, Perth, Western Australia, Australia, 2 Faculty of Health Science, Curtin School of Population Health, Curtin University, Bentley, Western Australia, Australia, 3 Health and Clinical Analytics, School of Public Health, University of Sydney, Sydney, New South Wales, Australia, 4 Northern Clinical School, University of Sydney, Sydney, New South Wales, Australia, 5 Women and Babies Health Research, Kolling Institute, Northern Sydney Local Health District, Sydney, New South Wales, Australia, 6 National Centre for Immunisation Research and Surveillance of Vaccine Preventable Diseases, The Children's Hospital at Westmead, Sydney, New South Wales, Australia, 7 School of Public Health and Community Medicine, University of New South Wales Medicine, Sydney, New South Wales, Australia, 8 Division of Pediatrics, University of Western Australia, Perth, Western Australia, Australia

* tom.snelling@sydney.edu.au

## Abstract

### Background

Asthma is among the commonest noncommunicable diseases of childhood and often occurs with other atopic comorbidities. A previous case-control study found evidence that compared to children who received acellular pertussis (aP) vaccines in early infancy, children who received one or more doses of whole-cell pertussis (wP) vaccine had lower risk of developing IgE-mediated food allergy. We hypothesized that wP vaccination in early infancy might protect against atopic asthma in childhood.

### Methods

Retrospective record-linkage cohort study of children between 5 and < 15 years old and born between January 1997, and December 1999, in the Australian states of Western Australia (WA) and New South Wales (NSW), receiving wP versus aP vaccine as the first pertussis vaccine dose. The main outcome and measures were first and recurrent hospitalizations for asthma; hazard ratios (HRs) and 95% confidence intervals (CIs) were computed by means of Cox and Andersen and Gill models.

### Results

274,405 children aged between 5 and < 15 years old (78.4% NSW-born) received a first dose of either wP (67.8%) or aP vaccine before 4 months old. During the follow-up period, there were 5,905 hospitalizations for asthma among 3,955 children. The incidence rate for first hospitalization was 1.5 (95% CI 1.4–1.5) per 1,000 child-years among children receiving wP vaccine as a first dose, and 1.5 (95% CI 1.4–1.6) among those vaccinated with aP

This is a Registered Report and may have an associated publication; please check the article page on the journal site for any related articles.

**Data Availability Statement:** Linked administrative data are de-identified and are not owned by the authors. The custodians of the data used for this analysis are the New South Wales Centre for Health Record Linkage (https://www.cherel.org.au/) and Western Australia – Data Linkage (https://www.datalinkage-wa.org.au/). Interested researchers may apply at these sites for data access. Programming code for survival curves is available in the Supporting Information files.

**Funding:** This study was funded by the Population Health Research Network Proof of Concept Project, the Australian National Health and Medical Research Council (Chief Investigator Heather F Gidding APP1082342; https://www.nhmrc.gov.au/ ), and the Wesfarmers Centre of Vaccines and Infectious Diseases seed funding grant (round 1-2018, GPC, HFG, HCM, TS; https://infectiousdiseases.telethonkids.org.au/). GPC was funded by a Stan Perron Post-PhD Career Launching Award (2022; https://www.telethonkids.org.au/), the Australian Department of Education and Training Endeavour Scholarship (https://internationaleducation.gov.au/scholarships/Scholarships-and-Fellowships/Pages/default.aspx), and top-up scholarships from the Wesfarmers Centre of Vaccine and Infectious Diseases at the Telethon Kids Institute (https://infectiousdiseases.telethonkids.org.au/) and the Forrest Research Foundation (https://www.forrestresearch.org.au/). HCM and HCM were funded by the Australian National Health and Medical Research Council fellowships (https://www.nhmrc.gov.au/). TS is supported by a Medical Research Future Fund Investigator Grant (MRF1195153; https://www.health.gov.au/our-work/medical-research-future-fund).

**Competing interests:** "I have read the journal's policy and the authors of this manuscript have the following competing interests: No funding was received from commercial sources for this work. Associate Professor Moore is in receipt of research funds from Merck Sharp and Dohme (MSD) and Sanofi unrelated to the work presented in this paper. Associate Professor Moore has also received institutional honoraria for participating in advisory committees (Pfizer, MSD, Sanofi), also unrelated to the work presented in this paper. Associate Professor Gidding has received honoraria for participating in a Seqirus advisory committee unrelated to the work presented in this paper. Professor Richmond has served on pertussis vaccine scientific advisory boards for GlaxoSmithKline and Sanofi on behalf of his institution. He also participated in multicenter vaccine trials of pertussis vaccines sponsored by

vaccine as a first dose. The adjusted HRs for those who received wP vaccine versus aP vaccine as the first dose were 1.02 (95% CI 0.94–1.12) for first hospitalizations and 1.07 (95% CI 0.95–1.2) for recurrent hospitalizations for asthma.

## Conclusions

We found no convincing evidence of a clinically relevant association between receipt of wP versus aP vaccines in early infancy and hospital presentations for asthma in childhood.

## Introduction

Asthma is a heterogenous syndrome resulting from a complex interplay of genetic predisposition and environmental factors [1]. The type 2 'early-onset' asthma phenotype prevails in children and frequently co-occurs with other atopic comorbidities [2]. No effective prevention strategies have been identified [2].

The Australian National Health Survey (2007–2008) estimated that 10.4% (95% confidence interval [CI] 9.1% to 11.7%) of children under 15 years old had current asthma [3], a pattern that remained unaltered in the subsequent decade [4]. In 2015, the Australian health expenditure associated with asthma was $AU770 million, with at least one-third utilised in hospital services [5]. In the same setting, self-reported asthma was the leading cause of the total burden of disease due to respiratory illness in children aged between 5 and < 15 years old and the leading cause of non-fatal disease burden among those under 15 years old https://www.aihw.gov.au/reports/chronic-respiratory-conditions/chronic-respiratory-conditions/contents/asthma [5]. Uncontrolled childhood asthma is associated with sleep disturbances (49.7%; 95% CI 48.8% to 50.6%) and school absenteeism (46.3%; 95% CI 45.4% to 47.1%) [6]. Furthermore, early persistent asthma or persistent wheezing disorders have been associated with impaired lung development throughout childhood and adolescence, further impacting lung function in adulthood [7].

Between 1997 and 1999, Australia transitioned from using whole-cell pertussis (wP) vaccine to acellular pertussis (aP) vaccine for its 2, 4, and 6-month primary vaccine series [8], driven by the better tolerability profile of aP vaccine formulations [9]. The switchover from the wP to the aP regimen was chronologically overlapped with the transition from the ninth to the tenth edition of the International Classification of Diseases (ICD) and an increase in hospitalizations coded as food-associated anaphylaxis [10, 11]. A case-control study of children born during this period, found that those with allergist-diagnosed IgE-mediated food allergy were less likely than matched controls to have received wP vaccine as their first dose (odds ratio 0.77; 95% confidence interval [CI] 0.62–0.95) [12]. The findings raised the question of whether a similar association might also be observable for atopic asthma.

A previous immunological study suggests that an early first dose of wP predominantly stimulates a T helper (Th)$_1$ immune response, characterized by the secretion of *Bordetella pertussis'* epitope-specific interferon-gamma; by contrast, a first dose of aP vaccine elicits a skewed Th$_2$ immune responses and therefore, interleukin-5 polarizing signals [13]. This differential T cell polarization is maintained throughout adolescence and adulthood irrespective of subsequent booster doses of aP vaccine [13, 14]. We hypothesize that a dose of wP vaccine in early infancy could reduce the risk of IgE-mediated food allergy, because wP promotes the physiological shift of the young infants' Th$_2$-biased immunophenotype into a more balanced Th$_1$/Th$_2$/Th$_{17}$ immunophenotype [14]. Therefore, we designed this retrospective, record-linkage, population-based study primarily to determine whether a first dose of wP versus aP was associated

industry, also unrelated to the work presented in this paper. He has received no personal remuneration for these activities. No other disclosures were reported. This does not alter our adherence to PLOS ONE policies on sharing data and materials"

with a lower risk of asthma-related hospital presentations in childhood. We were interested in assessing protection against asthma in children aged between 5 and < 15 years old, because most wheezing episodes in younger children are driven by viral respiratory infections without upregulation of $Th_2$-associated pathways [15]. A secondary aim was to evaluate whether three primary doses of wP versus three primary doses aP, and at least one primary dose of wP versus three primary doses of aP, were associated with a decreased risk of hospitalizations for asthma.

## Methods

### Study setting and participants

We conducted a pre-registered, retrospective, population-based record-linkage cohort study to examine the association between the type of first pertussis vaccine dose received in infancy and hospitalizations with a primary diagnosis coded as asthma (as per the tenth edition of the International Classification of Diseases [ICD-10 AM] coding scheme) for children aged between 5 to <15 years old (protocol DOI: 10.1371/journal.pone.0260388) [16]. Details of the full cohort, datasets, data cleaning, and linkage procedures have been described elsewhere [17, 18]. Birth records were probabilistically linked to health data including immunization register and hospitalization data [17, 18]. The cohort was identified through the birth registries of the Australian states of New South Wales (NSW) and Western Australia (WA), and the NSW Perinatal Data Collection and WA Midwives' Notification System. The linkage accuracy between birth register and immunization datasets was 99.0%, and between the birth registry and death data 96.6% [17]. We included all children born between January 1, 1997, and December 31, 1999, who received a first dose of either wP or aP-containing vaccine before 4 months old, irrespective of subsequent doses, and had records in the perinatal and birth registries. Inclusion and exclusion criteria are operationalized in the S1 Table.

This study was approved by the human research ethics committees of the Department of Health of WA (approval number: 2012/75), NSW Population Health Service (approval number: HREC/13/CIPHS/15), Australian Institute of Health and Welfare (approval number: EC2012/4/62), Curtin University (approval number: HRE2019-0350), the WA Aboriginal Health Ethics Committee (approval number: 459), and the Aboriginal Health and Medical Research Council Ethics Committee (approval number: 931/13). A waiver of consent was requested and granted owing to the large size of the study cohort.

The data were supplied to the study researchers via the Secure Unified Research Environment (SURE; www.sure.org.au). By using the best practice methods for privacy preservation in linkage (the separation principle) the researchers did not have access to any identifiable information (including the names, addresses, and contact details of the participants).

### Exposures

The exposure of interest was the first dose of pertussis-containing vaccine (wP or aP) before 4 months old, irrespective of the vaccine type given for subsequent doses (primary analysis). Two additional exposure groups were defined and created by subsetting children with a three-dose primary pertussis vaccination series before the start of follow-up (secondary analyses). The first characterized the exposure as any doses of wP (with or without aP) versus aP-only doses (aP/aP/aP), and the second characterized the exposure as wP-only doses (wP/wP/wP) versus aP-only doses (aP/aP/aP). The vaccination status was ascertained from the Australian Immunisation Register (previously known as Australian Childhood Immunisation Register), a nationwide register that records all routine vaccinations given to children enrolled in Medicare, Australia's universal health insurance scheme.

## Outcomes

**Hospitalizations.** The outcomes of interest were first and recurrent hospitalizations assigned a primary diagnosis of asthma per ICD-10 AM (S2 Table). Recurrent events were defined as those occurring at least 14 days after the previous episode. Hospitalizations were ascertained from the NSW Admitted Patient Data Collection and the WA Hospital Morbidity Data Collection. The hospitalization data included all discharges, transfers, and deaths in WA (from January 1996) and NSW (from July 2001) [17].

**Emergency department presentations.** First and recurrent presentations to the emergency department for asthma were also examined as study outcomes. Recurrent events were defined as above.

For WA-born children, emergency department presentations were ascertained from the WA Emergency Department Data Collection using the principal diagnosis (ICD-10 AM diagnostic codes), symptom codes (Systematized Nomenclature of Medicine, Clinical Terms, SNOMED-CT; S2 Table), diagnosis at discharge (free-text), and presenting complaint (free-text). The dataset includes presentations to all hospitals from 2002 until 2013 [17]. These outcomes were determined using a hierarchical rule, in which more specific diagnostic categories were chosen in preference to the less specific categories, depending on their availability [19].

For NSW-born children, asthma presentations to the emergency department were ascertained from the NSW Emergency Department Data Collection, with diagnoses coded using SNOMED-CT and ICD-10 AM (S2 Table). While the dataset includes presentations to nearly all metropolitan public hospital emergency departments in NSW from 2005 until 2013, the proportion of hospitals contributing data varied each year [17].

## Confounders

Potential confounders and prognostically important covariates present at birth (prior to primary vaccination) were identified using a preliminary causal model represented as a directed acyclic graph (DAG) [16]. Residual confounding was assessed using hospitalizations for negative control outcomes, namely an ICD-10 AM coded primary diagnosis relating to injury, trauma, or poisoning (S3 Table). The model assumptions for the assessment of unmeasured confounding and relevant operational definitions are presented in the S1 Fig and S4 Table.

## Statistical analysis

We separately analyzed (1) hospitalizations for asthma in children born in NSW or WA; (2) presentations to the emergency department for asthma in the NSW cohort and (3) presentations to the emergency department for asthma in the WA cohort. Time-to-first event and time-to-recurrent events analyses were conducted using Cox and Andersen-Gill regression models, respectively, with robust variance estimates to account for potential within-child dependencies [20–23]. Operational definitions are provided in the S5 Table.

Owing to differences in the availability of the outcome data, the person-time of follow-up started at 5 years old for the analyses of hospitalizations, and at 5 and 8 years old for presentations to the emergency department in the WA and NSW cohorts, respectively. For the Cox models, the time-to-first hospitalization with a primary asthma-related ICD-10 AM diagnostic code was the event of interest. For the Andersen-Gill models, we considered events occurring at least 14 days after the previous episode as a recurrence. For all analyses, the data were censored on the child's 15th birthday, the end of the study period (December 31, 2013), or death, whichever was the earliest. Primary and secondary complete-case analyses (i.e., including only observations with no missing data) were performed to estimate unadjusted and adjusted hazard ratios (HRs), and their 95% CIs [24]. Confounders and prognostically important covariates

were selected a priori based on causal considerations, a prespecified DAG, and the modified disjunctive cause criterion [16, 25]. Statistical models were stratified by state of birth, allowing for different baseline hazards, and adjusted for socioeconomic status [26], remoteness, year and season of birth, birth order, maternal smoking during pregnancy, method of delivery, gestational age at delivery, Aboriginal status, and sex, as recorded on the birth registries and perinatal data collections, and measured before the administration of the first dose of wP/aP. To avoid imposing a linear relationship between gestational age at delivery and the outcomes, we specified this continuous variable as a penalized spline [22, 27]. with optimal degrees of freedom chosen based on the Akaike Information Criterion.

To explore age-dependent modification of the effect of the first dose of pertussis-containing vaccine, we assessed the interaction between the vaccine type (first dose) and the age of administration of that dose with a cut point at 3 months old. These models were compared using a likelihood ratio test. Pre-specified sensitivity analyses on the case definition of asthma were conducted by restricting cases to just those with the primary diagnosis code of 'predominantly allergic asthma' (ICD 10-AM: J45.0).

This study is reported as per the REporting of studies Conducted using Observational Routinely-collected Data (RECORD) statement (S1 Checklist: RECORD Checklist) [28]. The cohort was curated in Stata version 16.0; [29] further data manipulation and statistical analyses were performed in R version 4.0.3.19 [30], using the tidyverse core packages [31], the survival and survminer libraries [22, 32].

## Bias

Differential likelihood of surviving to cohort entry among children receiving a first dose of wP versus aP was identified as a potential source of selection bias. However, in the Australian context, infant and child mortality rates are exceptionally low and deaths from pertussis or atopic diseases are extremely rare. A difference in the causal effect of infant vaccination on asthma among younger and older children was identified as another potential source of bias, such that any effect measured in older children only could be different from the overall effect. Additional potential sources of bias are detailed elsewhere [16].

## Results

### Association between pertussis vaccination and hospitalizations for asthma

Between January 1, 1997, and December 31, 1999, there were 329,831 births with records in both the state-specific birth registry and perinatal databases. After exclusions (Fig 1), the cohort comprised 283,440 children, of whom 192,644 (68.0%) received wP as their first pertussis vaccine dose, 269,011 (94.9%) received a three-dose primary pertussis series by the date of the 5th birthday, and 207,090 (73.1%) received homologous priming with three doses of wP or three doses of an aP-based vaccine formulation. Birth in 1997 or 1998 were more common among children who received a first dose of wP versus aP (Table 1), consistent with the progressive transition from using wP to aP over those years.

During the follow-up period, 4,073 (1.4%) of the 283,440 children had hospitalizations for asthma-related ICD codes leading to 6,102 hospitalizations. Of these 4,073 children, 1,559 (38.3%) were female, 220 (5.4%) were Aboriginal, 3,174 (77.9%) were born in NSW, and 2,775 (68.1%) received wP as their first pertussis vaccine dose. 274,405 (96.8%) of 283,440 had complete-case data and were included in the primary analysis (Table 2). Of them, 78.4% were NSW-born and 67.8% received wP as their first pertussis vaccine dose (S6 Table).

The incidence rate of first hospitalization assigned a principal diagnosis of asthma was 1.5 (95% CI 1.4–1.5) and 1.5 (95% CI 1.4–1.6) per 1,000 child-years among those vaccinated with

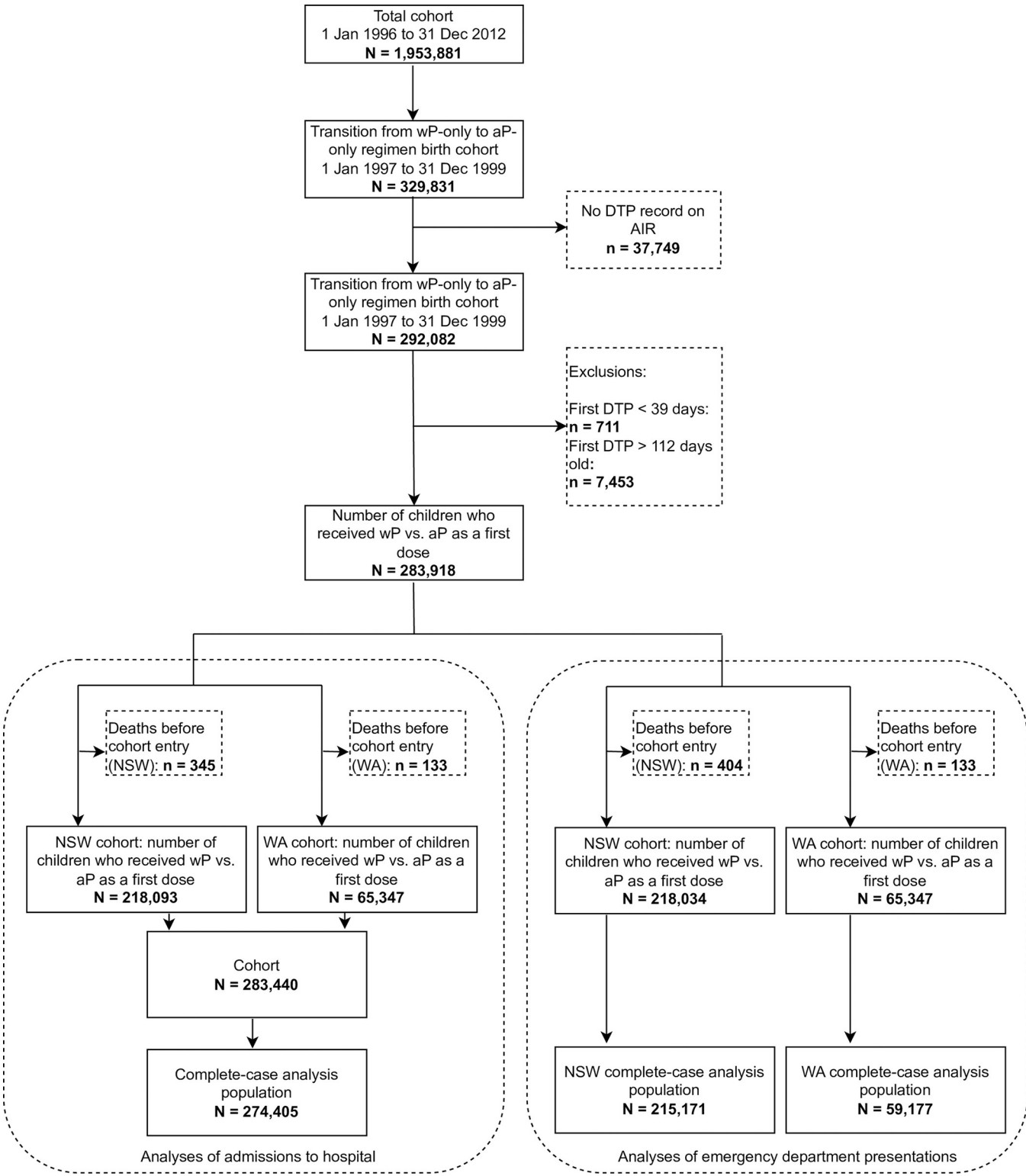

**Fig 1. Flow chart of cohort ascertainment and exclusions.** Jan, January; Dec, December; DTP, diphtheria-tetanus-pertussis vaccine; AIR, Australian Immunisation Register; NSW, New South Wales; WA, Western Australia.

**Table 1. Characteristics of the cohort (children born in New South Wales or Western Australia between 1997 and 1999) by state of birth and type of first dose of pertussis-containing vaccine.**

| | wP (n = 192,644) | | aP (n = 90,796) | |
|---|---|---|---|---|
| Characteristic | NSW (n = 148,410) | WA (n = 44,234) | NSW (n = 69,683) | WA (n = 21,113) |
| **Maternal age (years)** | | | | |
| Median (Q1-Q3) | 29 (25–33) | 29 (25–32) | 30 (26–33) | 29 (26–33) |
| **Number of previous pregnancies—No. (%)** | | | | |
| 0 | 59,300 (39.9) | 12,349 (27.9) | 30,793 (44.2) | 6,884 (32.6) |
| 1 | 51,731 (34.9) | 13,930 (31.5) | 23,588 (33.9) | 6,692 (31.7) |
| 2 | 23,914 (16.1) | 8,777 (19.8) | 10,185 (14.6) | 3,796 (18.0) |
| ≥ 3 | 13,465 (9.1) | 9,178 (20.7) | 5,117 (7.3) | 3,741 (17.7) |
| **Maternal smoking in pregnancy—No. (%)** | | | | |
| No | 119,624 (80.6) | 37,216 (84.1) | 58,727 (84.3) | 17,464 (82.7) |
| Yes | 28593 (19.3) | 7,018 (15.9) | 10,938 (15.7) | 3,649 (17.3) |
| Unknown | 193 (0.1) | NA | 18 (0.0) | NA |
| **Mother born overseas—No. (%)** | | | | |
| Australia | 109,013 (73.4) | 870 (2.0) | 52,747 (75.7) | 295 (1.4) |
| Overseas | 39,148 (26.4) | 258 (0.6) | 16,904 (24.3) | 116 (0.5) |
| Missing | 249 (0.2) | 43,106 (97.4) | 32 (0.0) | 20,702 (98.1) |
| **Paternal age (years)** | | | | |
| Median (Q1-Q3) | 32 (28–36) | 31 (27–35) | 32 (28–36) | 32 (28–36) |
| **Socioeconomic index—No. (%)** | | | | |
| 91 to 100% (least disadvantaged) | 10,352 (7.0) | 2,664 (6.0) | 7,472 (10.7) | 1,783 (8.4) |
| 76 to 90% | 18,499 (12.5) | 4,797 (10.8) | 11,287 (16.2) | 2,996 (14.2) |
| 26 to 75% | 73,625 (49.6) | 20,062 (45.4) | 33,309 (47.8) | 10,086 (47.7) |
| 11 to 25% | 25,275 (17.0) | 7,596 (17.2) | 9,777 (14.0) | 3,305 (15.7) |
| 0 to 10% (most disadvantaged) | 19,326 (13.0) | 4,483 (10.1) | 7,036 (10.1) | 1,411 (6.7) |
| Unknown | 1,333 (0.9) | 4,632 (10.5) | 802 (1.2) | 1,532 (7.3) |
| **Accessibility or remoteness index of Australia—No. (%)** | | | | |
| Major cities | 109,118 (73.5) | 26,825 (60.6) | 52,133 (74.8) | 16,009 (75.8) |
| Inner and outer regional | 37,335 (25.1) | 9,074 (20.5) | 16,450 (23.6) | 3,160 (14.9) |
| Remote and very remote | 981 (0.7) | 3,703 (8.4) | 457 (0.7) | 412 (2.0) |
| Unknown | 976 (0.7) | 4,632 (10.5) | 643 (0.9) | 1,532 (7.3) |
| **Sex—No. (%)** | | | | |
| Female | 72,148 (48.6) | 21,586 (48.8) | 34,026 (48.8) | 10,406 (49.2) |
| **Year of birth—No. (%)** | | | | |
| 1997 | 69,210 (46.6) | 19,160 (43.3) | 2,728 (3.9) | 1,665 (7.9) |
| 1998 | 61,234 (41.3) | 17,126 (38.7) | 11,172 (16.0) | 5,022 (23.8) |
| 1999 | 17,966 (12.1) | 7,948 (18.0) | 55,783 (80.1) | 14,426 (68.3) |
| **Aboriginal status—No. (%)** | | | | |
| Non- Aboriginal | 140,674 (94.8) | 41,310 (93.4) | 66,649 (95.6) | 20,547 (97.3) |
| Aboriginal and/or Torres Strait Islander | 7,340 (4.9) | 2,924 (6.6) | 3,027 (4.3) | 566 (2.7) |
| Unknown | 396 (0.3) | NA | 7 (0.0) | NA |
| **Gestational age at birth (weeks)** | | | | |
| Median (Q1-Q3) | 40 (38–40) | 39 (38–40) | 40 (38–40) | 39 (38–40) |
| **Delivery method—No. (%)** | | | | |
| Vaginal | 104,673 (70.5) | 27,980 (63.2) | 46,400 (66.6) | 12,227 (57.9) |
| Instrumentation | 15,504 (10.4) | 6090 (13.8) | ≥ 6 | 3,377 (16.0) |
| Caesarean | 28,153 (19.0) | 10,164 (23.0) | 14,729 (21.1) | 5,509 (26.1) |

*(Continued)*

**Table 1.** (Continued)

| Characteristic | wP (n = 192,644) | | aP (n = 90,796) | |
|---|---|---|---|---|
| | NSW (n = 148,410) | WA (n = 44,234) | NSW (n = 69,683) | WA (n = 21,113) |
| Unknown | 80 (0.1) | NA | < 6 | NA |
| **Apgar (5 min)—No. (%)** | | | | |
| 10 | 32,521 (21.9) | 10,419 (23.5) | 15,205 (21.8) | 4,505 (21.3) |
| 9 | 103,376 (69.6) | 30,033 (67.9) | 48,501 (69.6) | 14,721 (69.7) |
| 8 | 7,499 (5.1) | 2,565 (5.8) | 3,563 (5.1) | 1,268 (6.0) |
| $\leq$ 7 | 4,756 (3.2) | 1,174 (2.7) | 2,315 (3.3) | 598 (2.8) |
| Unknown | 258 (0.2) | 43 (0.1) | 99 (0.1) | 21 (0.1) |
| **Birth weight—grams** | | | | |
| Median (Q1-Q3) | 3,420 (3,080–3,750) | 3,395 (3,060–3,725) | 3,420 (3,085–3,760) | 3,400 (3,060–3,730) |
| **Season of birth—No. (%)** | | | | |
| Spring | 34,956 (23.6) | 10,371 (23.4) | 20,574 (29.5) | 5,990 (28.4) |
| Winter | 36,136 (24.3) | 10,797 (24.4) | 19,355 (27.8) | 5,602 (26.5) |
| Autumn | 38,117 (25.7) | 11,600 (26.2) | 16,401 (23.5) | 5,126 (24.3) |
| Summer | 39,201 (26.4) | 11,466 (25.9) | 13,353 (19.2) | 4,395 (20.8) |
| **Age at first dose of wP/aP (days)** | | | | |
| Median (Q1-Q3) | 62 (58–67) | 60 (57–66) | 61 (58–67) | 60 (56–64) |
| **Age at first dose of wP/aP (days)—No. (%)** | | | | |
| $\geq$ 39 and < 91 | 144,440 (97.3) | 43,084 (97.4) | 67,914 (97.5) | 20,668 (97.4) |
| $\geq$ 91 and < 112 | 3,970 (2.7) | 1,150 (2.6) | 1,769 (2.5) | 445 (2.6) |
| **At least one hospitalization ICD-coded as asthma from 5 years old—No. (%)** | | | | |
| Yes | 2,135 (1.4) | 640 (1.4) | 1,039 (1.5) | 259 (1.2) |

Abbreviations: wP, whole-cell pertussis vaccine; aP, acellular pertussis vaccine; NSW, New South Wales; WA, Western Australia; NA, not applicable; Q1, first quartile; Q3, third quartile. Frequency counts that represent < 6 children were suppressed, per the Australian Institute of Health and Welfare curation guidelines as of March 2022. Proportions may not add to 100% due to rounding

wP versus aP as their first dose, respectively; the incidence rate for recurrent hospitalizations with a principal diagnosis coded as asthma was 2.2 (95% CI 2.1–2.3) in wP and aP vaccinated children.

Fig 2 shows unadjusted survival curves for up to 10 years of follow-up for children who received wP versus aP as their first dose. Adjusted survival curves by vaccination status and stratified by state of birth are shown in the S2 Fig. Compared with children receiving aP as their first dose, the adjusted HRs for time-to-first hospitalization and time-to-recurrent hospitalizations in children vaccinated with a first dose of wP were 1.02 (95% CI 0.94–1.12) and 1.07 (95% CI 0.95–1.2), respectively. The likelihood ratio test showed no evidence of modification of the effect of the first primary dose of wP versus aP by the age at administration (Table 2). Recurrent events, incidence rates and HRs for the remaining exposure groups, and sensitivity analyses are detailed in the S7 and S8 Tables and in Table 2.

## Association between pertussis vaccination and emergency department presentations for asthma

**NSW cohort.** After exclusions (Fig 1), the study population comprised 218,034 children born in NSW between 1997 and 1999, who received their first pertussis-containing vaccine dose before 4 months old. During the follow-up period (i.e., from 8 to < 15 years old), 3,640 (1.7%) of 218,034 children had at least one presentation to the emergency department for

**Table 2. Hazard ratios and 95% confidence intervals for hospitalizations for asthma.**

| Exposure—Outcome | Analysis population (N)[a] | Hospitalizations (n) | Incidence rate (95% CI) per 1,000 child-years | Unadjusted HR (95% CI)[b] | Adjusted HR (95% CI)[c] |
|---|---|---|---|---|---|
| **wP versus aP as a first dose—Time-to-first hospitalization** | | | | | |
| aP | 88,424 | 1,267 | 1.5 (1.4–1.6) | 1 [Reference] | 1 [Reference] |
| wP | 185,981 | 2,688 | 1.5 (1.4–1.5) | 1.00 (0.94–1.07) | 1.02 (0.94–1.12)[d] |
| **wP versus aP as a first dose—Time-to-recurrent hospitalizations** | | | | | |
| aP | 88,424 | 1,846 | 2.2 (2.1–2.3) | 1 [Reference] | 1 [Reference] |
| wP | 185,981 | 4,059 | 2.2 (2.1–2.3) | 1.04 (0.96–1.14) | 1.07 (0.95–1.2) |
| **wP versus aP as a first dose—Time-to-first hospitalization (sensitivity analysis—principal diagnosis coded as J45.0)[e]** | | | | | |
| aP | 89,508 | 10 | 0 (0–0)[f] | 1 [Reference] | 1 [Reference] |
| wP | 189,906 | 37 | 0 (0–0)[g] | 1.73 (0.86–3.48) | NA[h] |
| **Any wP versus all aP—Time-to-first hospitalization** | | | | | |
| aP/aP/aP | 70,947 | 1,023 | 1.5 (1.4–1.6) | 1 [Reference] | 1 [Reference] |
| Any wP | 189,479 | 2,700 | 1.4 (1.4–1.5) | 0.98 (0.92–1.06) | 1.00 (0.91–1.1) |
| **Any wP versus all aP—Time-to-recurrent hospitalizations** | | | | | |
| aP/aP/aP | 70,947 | 1,468 | 2.2 (2.1–2.3) | 1 [Reference] | 1 [Reference] |
| Any wP | 189,479 | 4,059 | 2.2 (2.1–2.2) | 1.04 (0.95–1.14) | 1.07 (0.96–1.21) |
| **All wP versus all aP—Time-to-first hospitalization** | | | | | |
| aP/aP/aP | 70,947 | 1,023 | 1.5 (1.4–1.6) | 1 [Reference] | 1 [Reference] |
| wP/wP/wP | 129,411 | 1,828 | 1.4 (1.4–1.5) | 0.97 (0.9–1.05) | 1.04 (0.92–1.18) |
| **All wP versus all aP—Time-to-recurrent hospitalizations** | | | | | |
| aP/aP/aP | 70,947 | 1,468 | 2.2 (2.1–2.3) | 1 [Reference] | 1 [Reference] |
| wP/wP/wP | 129,411 | 2,712 | 2.1 (2.0–2.2) | 1.01 (0.92–1.12) | 1.12 (0.96–1.31) |

Abbreviations: CI, confidence interval; HR, hazard ratio; wP, whole-cell pertussis vaccine; aP, acellular pertussis vaccine.

[a]The analysis population included only those without missing data (approximately 97%).

[b]Unadjusted HRs were calculated with all the eligible members of the cohort.

[c]Adjusted HRs were calculated with complete cases. The multivariable models were stratified by state of birth and adjusted for year of birth, birth order (using number of previous pregnancies as a surrogate), maternal smoking during pregnancy, socioeconomic status, the index of accessibility/remoteness of Australia, sex, Aboriginal status, delivery method, season of birth, and gestational age as a penalized spline.

[d]Likelihood ratio test (separate models including age at first dose of wP/aP with or without a product term between type of first dose of pertussis-containing vaccine received and age at first dose): P = 0.80.

[e]J45.0: predominantly allergic asthma.

[f]Incidence rate (95% CI) per 100,000 child-years: 1.2 (0.6–2.1).

[g]Incidence rate (95% CI) per 100,000 child-years: 2.0 (1.4–2.7).

[h]NA: not modeled owing to the sparsity of events.

asthma; of these 1,410 (38.7%) were female, 236 (6.5%) were Aboriginal, 2,534 (69.6%) lived in a major city, and 2,493 (68.5%) received wP as their first pertussis vaccine dose. 215,171 (98.7%) of 218,034 had complete-case data and were included in the analysis (Table 3 and S9 Table).

The incidence rate of the first presentation to the emergency department for asthma was 2.4 (95% CI 2.4–2.6) and 2.6 (95% CI 2.4–2.7) per 1,000 child-years among those vaccinated with wP versus aP as their first dose; the incidence rate for subsequent presentations was 3.8 (95% CI 3.6–3.9) and 3.6 (95% CI 3.5–3.8) in wP and aP vaccinated children, respectively.

Fig 3 shows unadjusted survival curves for up to 7 years of follow-up for children who received wP versus aP as their first dose. Compared with children receiving aP as their first dose, the adjusted HRs for time-to-first event and time-to-recurrent events in children vaccinated with a first dose of wP were 0.96 (95% CI 0.87–1.05) and 1.04 (95% CI 0.92–1.18),

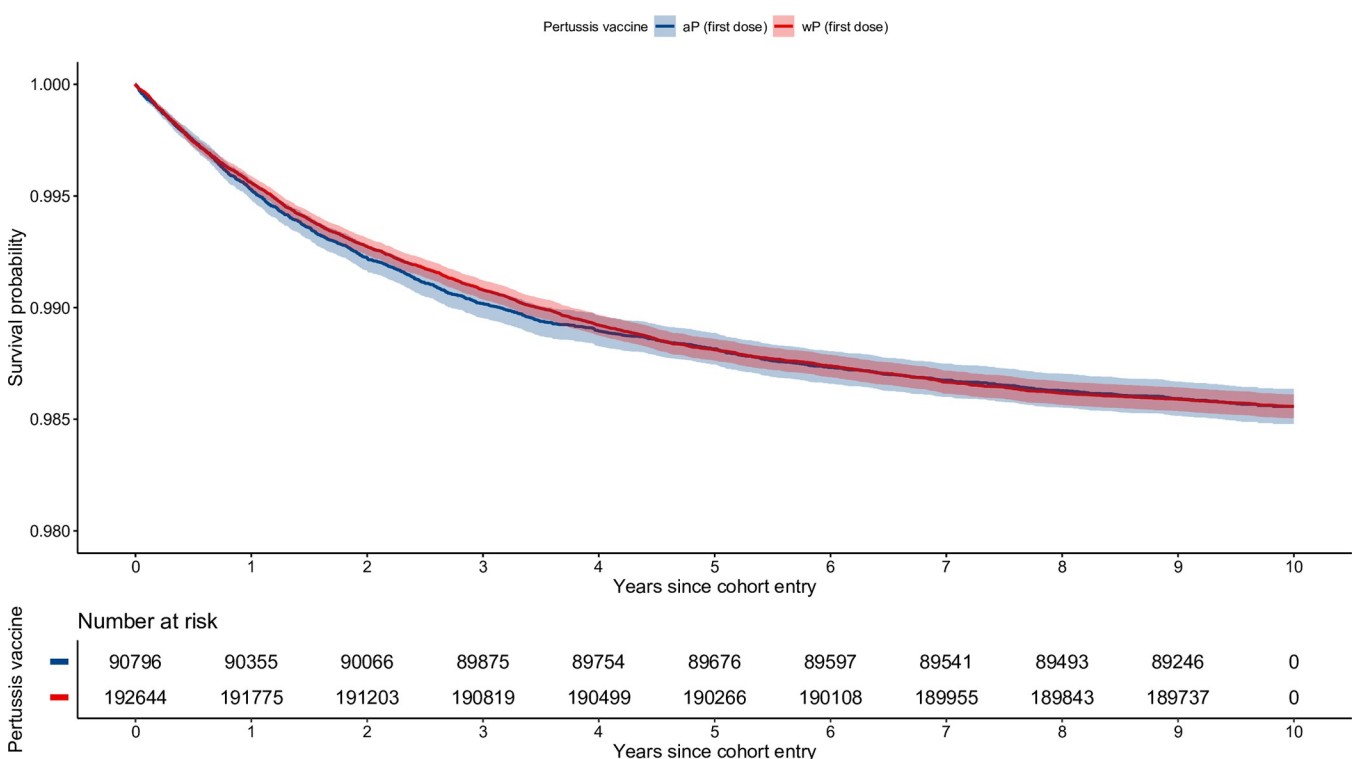

**Fig 2. Unadjusted survival curves.** Time-to-first hospitalization for asthma among children born in New South Wales or Western Australia between 1997 and 1999. wP as a first dose versus aP as a first dose. wP: whole-cell pertussis vaccine. aP: acellular pertussis vaccine. This figure was generated in R using the tidyverse core packages, as well as the survival and survminer libraries [22, 30–32].

respectively. Recurrent events, incidence rates, and HRs for the remaining exposure groups are detailed in the S10 and S11 Tables and in Table 3.

**WA cohort.** After exclusions (Fig 1), the study population comprised 65,347 children born in WA between 1997 and 1999, who received at least one dose of wP or aP before 4 months old. During the follow-up period (i.e., from 5 to < 15 years old), 740 (1.1%) of 65,347 children had at least one presentation to the emergency department due to asthma; of these 287 were female (38.8%), 20 were Aboriginal (2.7%), 551 lived in a major city (74.5%), and 503 received a first dose of wP (68.0%). 59,177 (90.6%) of 65,347 had complete-case data and were included in the analysis (Table 4 and S12 Table).

The incidence rate of the first presentation to the emergency department for asthma was 1.2 (95% CI 1.1–1.3) and 1.2 (95% CI 1.0–1.4) per 1,000 child-years among those vaccinated with wP versus aP as their first dose, respectively; the incidence rate for recurrent presentations was 1.8 (95% CI 1.7–1.9) and 1.5 (95% CI 1.4–1.7) in wP and aP vaccinated children, respectively.

Fig 4 shows unadjusted survival curves for up to 10 years of follow-up for children who received wP versus aP as their first dose. Compared with children receiving aP as their first dose, the adjusted HRs for time-to-first presentation and time-to-recurrent presentations in children vaccinated with a first dose of wP were 1.09 (95% CI 0.9–1.32) and 1.26 (95% CI 1.01–1.57), respectively. Recurrent events, incidence rates, HRs for the remaining exposure groups, and post-hoc sensitivity analyses of the NSW and WA cohorts using the date of the eighth birthday as cohort entry are detailed in Table 4 and in the S13–S15 Tables.

**Table 3. Hazard ratios and 95% confidence intervals for emergency department presentations for asthma in children born in New South Wales between 1997 and 1999.**

| Exposure–Outcome | Analysis population (N)[a] | Presentations (n) | Incidence rate (95% CI) per 1,000 child-years | Unadjusted HR (95% CI)[b] | Adjusted HR (95% CI)[c] |
|---|---|---|---|---|---|
| **wP versus aP as a first dose–Time-to-first presentation** | | | | | |
| aP | 68,822 | 1,143 | 2.6 (2.4–2.7) | 1 [Reference] | 1 [Reference] |
| wP | 146,349 | 2,472 | 2.4 (2.4–2.6) | 0.99 (0.92–1.06) | 0.96 (0.87–1.05) |
| **wP versus aP as a first dose–Time-to-recurrent presentations** | | | | | |
| aP | 68,822 | 1,645 | 3.6 (3.5–3.8) | 1 [Reference] | 1 [Reference] |
| wP | 146,349 | 3,828 | 3.8 (3.6–3.9) | 1.06 (0.97–1.16) | 1.04 (0.92–1.18) |
| **Any wP versus all aP–Time-to-first presentation** | | | | | |
| aP/aP/aP | 56,517 | 942 | 2.6 (2.4–2.8) | 1 [Reference] | 1 [Reference] |
| Any wP | 147,622 | 2,474 | 2.4 (2.4–2.5) | 0.98 (0.91–1.05) | 0.96 (0.87–1.07) |
| **Any wP versus all aP–Time-to-recurrent presentations** | | | | | |
| aP/aP/aP | 56,517 | 1,349 | 3.7 (3.5–3.9) | 1 [Reference] | 1 [Reference] |
| Any wP | 147,622 | 3,784 | 3.7 (3.6–3.8) | 1.04 (0.95–1.14) | 1.04 (0.91–1.17) |
| **All wP versus all aP–Time-to-first presentation** | | | | | |
| aP/aP/aP | 56,517 | 942 | 2.6 (2.4–2.8) | 1 [Reference] | 1 [Reference] |
| wP/wP/wP | 101,673 | 1,687 | 2.4 (2.3–2.5) | 0.93 (0.83–1.05) | 0.94 (0.81–1.09) |
| **All wP versus all aP–Time-to-recurrent presentations** | | | | | |
| aP/aP/aP | 56,517 | 1,349 | 3.7 (3.5–3.9) | 1 [Reference] | 1 [Reference] |
| wP/wP/wP | 101,673 | 2,583 | 3.6 (3.5–3.8) | 0.97 (0.83–1.12) | 1.04 (0.87–1.24) |

Abbreviations: CI, confidence interval; HR, hazard ratio; wP, whole-cell pertussis vaccine; aP, acellular pertussis vaccine.

[a]The analysis population included only those without missing data (approximately 99%).

[b]Unadjusted HRs were calculated with all the eligible members of the cohort.

[c]Adjusted HRs were calculated with complete cases. The multivariable models were adjusted for year of birth, birth order (using number of previous pregnancies as a surrogate), maternal smoking during pregnancy, socioeconomic status, the index of accessibility/remoteness of Australia, sex, Aboriginal status, delivery method, season of birth, and gestational age as a penalized spline.

### Negative control outcome analysis

The HR for time-to-first hospitalization coded as injury, trauma, or poisoning if vaccinated with a first dose of wP versus aP was 0.97 (95% CI 0.94–1.01). Incidence rates and HRs for the remaining exposure groups are detailed in the S16 Table.

## Discussion

In this population-based cohort study, we found little evidence of a clinically relevant difference in the risk of hospitalization or emergency department presentation for asthma among wP or aP-vaccinated children. There was evidence of a modestly increased rate of recurrent emergency department presentation for asthma among WA-born children vaccinated with a first dose of wP versus aP, but this was not observed for time-to-first emergency presentation, nor was it observed among the NSW cohort. Considering the totality of the evidence, we doubt this reflects a true difference in risk for wP and aP-vaccinated children born in WA. In our study, treatment effects were estimated using HRs among children receiving wP as a first dose versus aP as a first dose, at least one dose of wP versus aP-only doses, and wP-only primary doses versus aP-only primary doses. A known limitation of the Cox model for the effect of medical interventions on failure time, is that hazards and HRs are not (and are not expected to be) constant over time. For these analyses and the causal question of interest, the HRs of our Cox model can only be interpreted as a weighted average of the true hazard ratios over the

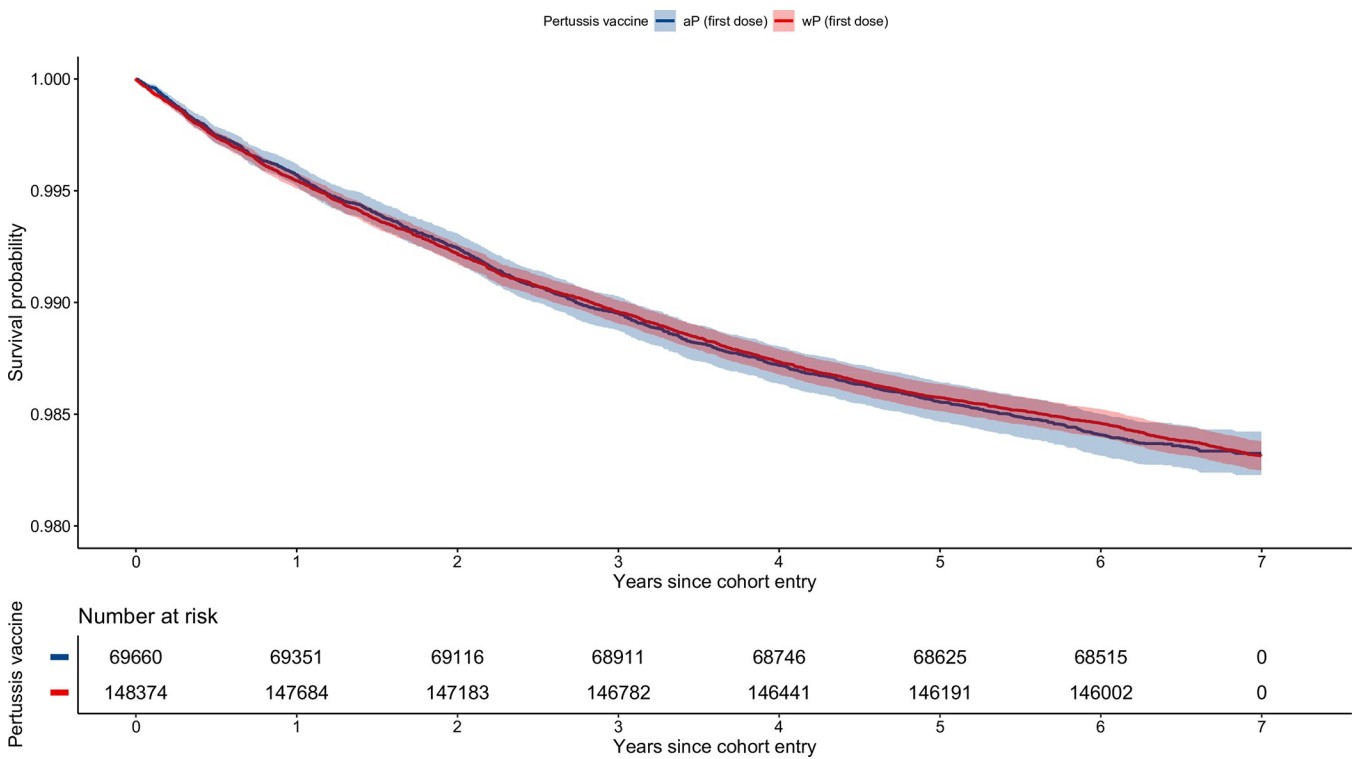

**Fig 3. Unadjusted survival curves.** Time-to-first presentation to the Emergency Department for asthma, among children born in New South Wales between 1997 and 1999. wP as a first dose versus aP as a first dose. wP: whole-cell pertussis vaccine. aP: acellular pertussis vaccine. This figure was generated in R using the tidyverse core packages, as well as the survival and survminer libraries [22, 30–32].

entire follow-up period [33]. Alternative statistical approaches such as computing the restricted mean survival time in lieu of HRs or bootstrapping 95% CI are beyond the scope of this analysis and could be explored by future studies [33]. Nonetheless, our study has several strengths. First, during the period of transition to the aP-only regimen, receipt of the first or subsequent doses as wP rather than aP vaccine was chiefly determined by chance, based on whichever vaccine was available at the vaccination provider on the date of vaccination. The vaccine availability was affected by the date of vaccination (and therefore by the date of birth) and presumably by the logistics of the rollout of the new aP vaccine formulation, which were likely to vary within and between jurisdictions. In other words, there is no reason to expect that parents or providers had any choice in which vaccine was delivered, and therefore confounding by risk factors for atopic outcomes is unlikely. Second, statistical models were based on DAGs [16]. The baseline characteristics of the cohort at birth that might otherwise confound the association between vaccination and hospital presentation for asthma were ascertained and adjusted for in the regression models, together with predictors of asthma that were not likely to have influenced the type of vaccine received as the first dose.

Former clinical studies suggest that early doses of wP might have a trophic effect on the maturation of the infant immune system [34, 35]. We have previously presented evidence that children who received wP rather than aP as their first dose may be partly protected against IgE-mediated food allergy [12]. We, therefore, speculated that a first dose of wP compared to aP before 4 months old might help avert the genesis of type 2 lung inflammation in the early postnatal life leading to subsequent development of early-onset asthma. However, the pathobiological processes that underpin these outcomes might already be underway prior to the first

**Table 4. Hazard ratios and 95% confidence intervals for emergency department presentations for asthma in children born in Western Australia between 1997 and 1999.**

| Exposure–Outcome | Analysis population (N)[a] | Presentations (n) | Incidence rate (95% CI) per 1,000 child-years | Unadjusted HR (95% CI)[b] | Adjusted HR (95% CI)[c] |
|---|---|---|---|---|---|
| **wP versus aP as a first dose–Time-to-first presentation** | | | | | |
| aP | 19,580 | 223 | 1.2 (1.0–1.4) | 1 [Reference] | 1 [Reference] |
| wP | 39,597 | 470 | 1.2 (1.1–1.3) | 1.01 (0.87–1.18) | 1.09 (0.9–1.32) |
| **wP versus aP as a first dose–Time-to-recurrent presentations** | | | | | |
| aP | 19,580 | 291 | 1.5 (1.4–1.7) | 1 [Reference] | 1 [Reference] |
| wP | 39,597 | 703 | 1.8 (1.7–1.9) | 1.16 (0.95–1.4) | 1.26 (1.01–1.57) |
| **Any wP versus all aP–Time-to-first presentation** | | | | | |
| aP/aP/aP | 14,421 | 169 | 1.2 (1–1.4) | 1 [Reference] | 1 [Reference] |
| Any wP | 41,856 | 487 | 1.2 (1.1–1.3) | 0.95 (0.8–1.13) | 1.03 (0.84–1.27) |
| **Any wP versus all aP–Time-to-recurrent presentations** | | | | | |
| aP/aP/aP | 14,421 | 216 | 1.6 (1.4–1.8) | 1 [Reference] | 1 [Reference] |
| Any wP | 41,856 | 724 | 1.8 (1.6–1.9) | 1.09 (0.88–1.35) | 1.22 (0.95–1.57) |
| **All wP versus all aP–Time-to-first presentation** | | | | | |
| aP/aP/aP | 14,421 | 169 | 1.2 (1–1.4) | 1 [Reference] | 1 [Reference] |
| wP/wP/wP | 27,716 | 319 | 1.2 (1–1.3) | 0.94 (0.78–1.12) | 1.1 (0.85–1.43) |
| **All wP versus all aP–Time-to-recurrent presentations** | | | | | |
| aP/aP/aP | 14,421 | 216 | 1.6 (1.4–1.8) | 1 [Reference] | 1 [Reference] |
| wP/wP/wP | 27,716 | 455 | 1.6 (1.6–1.8) | 1.03 (0.83–1.29) | 1.25 (0.92–1.71) |

Abbreviations: CI, confidence interval; HR, hazard ratio; wP, whole-cell pertussis vaccine; aP, acellular pertussis vaccine.

[a]The analysis population included only those without missing data (approximately 91%).

[b]Unadjusted HRs were calculated with all the eligible members of the cohort.

[c]Adjusted HRs were calculated with complete cases. The multivariable models were adjusted for year of birth, birth order (using number of previous pregnancies as a surrogate), maternal smoking during pregnancy, socioeconomic status, the index of accessibility/remoteness of Australia, sex, Aboriginal status, delivery method, season of birth, and gestational age as a penalized spline.

pertussis vaccine dose. Therefore, a first dose of wP given at approximately 2 months old may fall beyond the yet unknown window of opportunity for preventing asthma-related hospitalizations (a proxy of severe asthma exacerbations) in children aged 5 years or older.

A previous randomized controlled trial of Swedish-born infants found no evidence of a difference in the cumulative incidence of symptoms of asthma by 7 years old across the study arms; the crude relative risk (RR), if primed with wP versus diphtheria-tetanus (DT) toxoid vaccine, was 0.90 (95% CI 0.4–1.8) and 1.07 (95% CI 0.6–2.0) if primed with 5-component-DTaP versus DT [36]. Retrospective studies that looked at pertussis vaccination and subsequent development of asthma support no association or have been inconclusive [37–41].

The observational nature of the study means residual confounding is very possible, although the negative control outcome analyses found no evidence of it. We observed no evidence of modification of the effect of the first primary dose of wP versus aP by the age at administration, but only 2.7% of children received a delayed first dose of pertussis-containing vaccines. Other limitations of our approach are inherent in the method of ascertainment of the outcomes of interest. It is likely that hospitalizations and emergency department presentations due to acute asthma exacerbations were missed due to misclassification resulting in underascertainment of the outcome.

In conclusion, despite their putative $Th_1/Th_{17}$ polarizing properties, we found no conclusive evidence on an effect of wP versus aP vaccination in early infancy on hospitalizations and emergency department presentations for asthma in childhood.

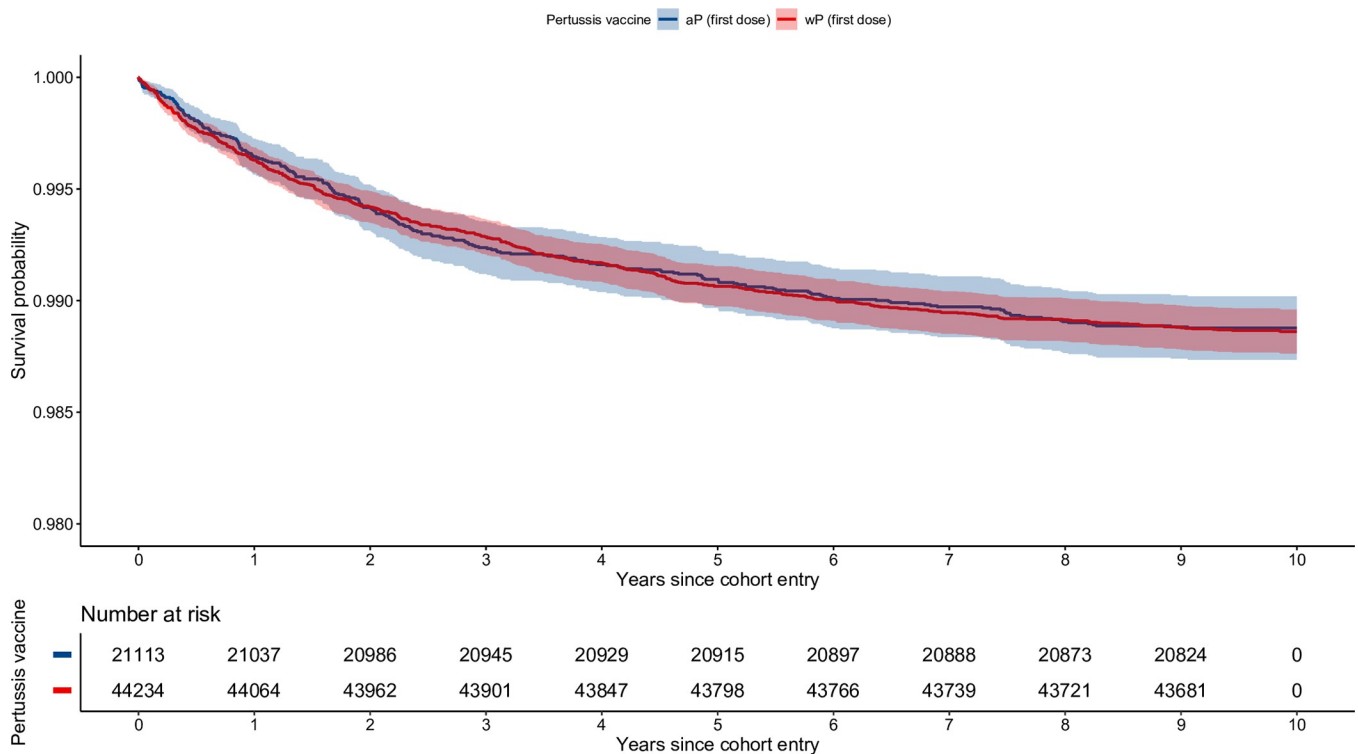

**Fig 4. Unadjusted survival curves.** Time-to-first presentation to the Emergency Department for asthma, among children born in Western Australia between 1997 and 1999. wP as a first dose versus aP as a first dose. wP: whole-cell pertussis vaccine. aP: acellular pertussis vaccine. This figure was generated in R using the tidyverse core packages, as well as the survival and survminer libraries [22, 30–32].

## Supporting information

**S1 Checklist. RECORD checklist.**
(PDF)

**S1 Table. Eligibility criteria.**
(PDF)

**S2 Table. Study outcomes.**
(PDF)

**S3 Table. Negative control outcomes.**
(PDF)

**S4 Table. Variable dictionary–directed acyclic graph.**
(PDF)

**S5 Table. Operational definitions.**
(PDF)

**S6 Table. Recurrent hospitalizations for asthma among children who received their first pertussis-containing vaccine dose before 4 months old.**
(PDF)

**S7 Table. Recurrent hospitalizations for asthma among children vaccinated with a three-dose primary pertussis vaccination series (i.e., any dose of wP versus aP-only doses) before**

cohort entry (i.e., 5 years old).
(PDF)

**S8 Table. Recurrent hospitalizations for asthma among children vaccinated with a three-dose primary pertussis vaccination series (i.e., wP-only doses versus aP-only doses) before cohort entry (i.e., 5 years old).**
(PDF)

**S9 Table. NSW cohort–recurrent presentations to the emergency department for asthma among children who received their first pertussis-containing vaccine dose before 4 months old.**
(PDF)

**S10 Table. NSW cohort–recurrent presentations to the emergency department for asthma among children receiving a three-dose primary pertussis vaccination series (i.e., any dose of wP versus aP-only doses) before cohort entry (i.e., 8 years old).**
(PDF)

**S11 Table. NSW cohort–recurrent presentations to the emergency department for asthma among children receiving a three-dose primary pertussis vaccination series (i.e., wP-only doses versus aP-only doses) before cohort entry (i.e., 8 years old).**
(PDF)

**S12 Table. WA cohort–recurrent presentations to the emergency department for asthma among children receiving at least one dose of pertussis-containing vaccine before 4 months old.**
(PDF)

**S13 Table. WA cohort–recurrent presentations to the emergency department for asthma among children receiving three-dose primary pertussis vaccination series (i.e., any dose of wP versus aP-only doses) before cohort entry (i.e., 5 years old).**
(PDF)

**S14 Table. WA cohort–recurrent presentations to the emergency department for asthma among children receiving a three-dose primary pertussis vaccination series (i.e., wP-only doses versus aP-only doses) before cohort entry (i.e., 5 years old).**
(PDF)

**S15 Table. NSW and WA cohorts–hazard ratios and 95% confidence intervals for emergency department presentations for asthma (cohort entry: 8 years old).**
(PDF)

**S16 Table. Hazard ratios and 95% confidence intervals for hospitalizations for injury, trauma, or poisoning.**
(PDF)

**S1 Fig. Directed acyclic graph.** Simplified directed acyclic graph describing the proposed causal relationship between pertussis immunization and admissions to hospital with an assigned principal diagnosis of asthma according to the International Classification of Diseases (9th edition, Clinical Modification or 10th edition, Australian Modification), in children born in Western Australia ($L21 = 0$) or New South Wales ($L21 = 1$) between 1997 and 1999 *(L20)*. Admissions to hospital with an assigned principal diagnosis of injury, trauma, or poisoning per the same coding schemes represent the negative control outcome of this study *(Y6)*. These outcomes are not in the causal pathway between the exposure of interest *(A7)* and admissions for asthma (*Y18* or its descendant *Y19*), despite sharing a set of common causes.
Figure generated via GeNIe Academic version 4.0.1922.0.[1]. [1]BayesFusion. GeNIe Modeler

[Internet]. 2022. Available from: https://www.bayesfusion.com/genie/ .
(TIF)

**S2 Fig. Adjusted survival curves.** (A) Time-to-first hospitalization for asthma among children born in New South Wales or Western Australia between 1997 and 1999 stratified by state of birth. wP as a first dose versus aP as a first dose. wP: whole-cell pertussis vaccine. aP: acellular pertussis vaccine. (B) Number at risk table. This figure was generated in R using the tidyverse core packages, as well as the survival and survminer libraries [1–4]. [1]R Core Team. R: A Language and Environment for Statistical Computing [Internet]. Vienna, Austria: R Foundation for Statistical Computing; 2022. Available from: https://www.R-project.org/. [2]Wickham H, Averick M, Bryan J, Chang W, D'Agostino McGowan L, François R, Müller K et al. Welcome to the tidyverse. J. Open Source Softw. 2019; 4(43), 1686. doi:10.21105/joss.01686. [3]Therneau T. A package for survival analysis in R [Internet]. R package version 3.5–5. Available from: https://CRAN.R-project.org/package=survival. [4]Kassambara A, Kosinski M, Przemyslaw B. survminer: Drawing Survival Curves using 'ggplot2'. R package version 0.4.9. [Internet]. 2021. Available from: https://CRAN.R-project.org/package=survminer .
(ZIP)

**S1 File. R code, survival curves.**
(ZIP)

# Acknowledgments

We acknowledge Aboriginal and Torres Strait Islander People as the Traditional Custodians of the land and waters of Australia. We also acknowledge the Nyoongar Wadjuk, Yawuru, Kariyarra and Kaurna Elders, their people and their land upon which the Telethon Kids Institute is located, as well as the Gadigal people of the Eora Nation and their ancestral lands upon which the University of Sydney is built. We seek their wisdom in our work to improve the health and development of all children.

We thank the staff at the Population Health Research Network data linkage and infrastructure nodes (the WA Data Linkage Branch, the NSW Centre for Health Record Linkage, and the Australian Institute for Health and Welfare) and the WA and Commonwealth Departments of Health and NSW Ministry of Health who provided advice and the data. We would also like to thank the Wesfarmers Centre of Vaccines and Infectious Diseases Community Reference Group for their valuable insights and Dr Rosanne Barnes for her input into the study protocol. Our data sources are acknowledged below:

1. Perinatal data: NSW Perinatal Data Collection and WA Midwives Notification System. Births: NSW Birth Registration Data Collection and WA Registry of Births Deaths and Marriage.

2. Death data: National Death Index.

3. Immunization data: the AIR dataset.

4. Hospitalization data: NSW Admitted Patient Data Collection and WA Hospital Morbidity Data Collection.

5. Emergency department data: Emergency Department Data Collection (NSW and WA).

# Author Contributions

**Conceptualization:** Gladymar Pérez Chacón, Mark Jones, Peter C. Richmond, Tom Snelling.

**Data curation:** Gladymar Pérez Chacón, Parveen Fathima.

**Formal analysis:** Gladymar Pérez Chacón.

**Funding acquisition:** Gladymar Pérez Chacón, Heather F. Gidding, Hannah C. Moore, Tom Snelling.

**Investigation:** Gladymar Pérez Chacón, Tom Snelling.

**Methodology:** Gladymar Pérez Chacón, Mark Jones, Tom Snelling.

**Supervision:** Mark Jones, Peter C. Richmond, Tom Snelling.

**Writing – original draft:** Gladymar Pérez Chacón.

**Writing – review & editing:** Gladymar Pérez Chacón, Parveen Fathima, Mark Jones, Marie J. Estcourt, Heather F. Gidding, Hannah C. Moore, Peter C. Richmond, Tom Snelling.

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
