## [Decision Letter · Decision Letter 0]

7 Jun 2023

PONE-D-23-11079Association between pertussis vaccination in infancy and asthma: a population-based record linkage cohort studyPLOS ONE

Dear Dr. Pérez Chacón,

Thank you for submitting your manuscript to PLOS ONE. After careful consideration, we feel that it has merit but does not fully meet PLOS ONE’s publication criteria as it currently stands. Therefore, we invite you to submit a revised version of the manuscript that addresses the points raised during the review process.

We look forward to receiving your revised manuscript.

Kind regards,

Dong Keon Yon, MD, FACAAI

Academic Editor

PLOS ONE

Journal Requirements:

2. In your cover letter, please confirm that the research you have described in your manuscript, including participant recruitment, data collection, modification, or processing, has not started and will not start until after your paper has been accepted to the journal (assuming data need to be collected or participants recruited specifically for your study). In order to proceed with your submission, you must provide confirmation.

"This study was funded by the Population Health Research Network Proof of Concept Project, the Australian National Health and Medical Research Council (NHMRC project grant GNT1082342, chief investigator HFG; https://www.nhmrc.gov.au/), and the Wesfarmers Centre of Vaccines and Infectious Diseases seed funding grant (round 1-2018, GPC, HFG, HCM, TS; https://infectiousdiseases.telethonkids.org.au/).

GPC is funded by a Stan Perron Post-PhD Career Launching Award (2022; https://www.perronfoundation.org.au/home), the Australian Department of Education and Training Endeavour Scholarship (https://internationaleducation.gov.au/scholarships/Scholarships-and-Fellowships/Pages/default.aspx), and top-up scholarships from the Wesfarmers Centre of Vaccine and Infectious Diseases at the Telethon Kids Institute (https://infectiousdiseases.telethonkids.org.au/) and the Forrest Research Foundation (https://www.forrestresearch.org.au/).

HCM and HCM were funded by the Australian National Health and Medical Research Council fellowships (https://www.nhmrc.gov.au/).

TS is supported by a Medical Research Future Fund Investigator Grant (MRF1195153; https://www.health.gov.au/our-work/medical-research-future-fund).

The funders did not have a role in study design, data collection and analysis, decision to publish, or preparation of the manuscript."

We note that one or more of the authors is affiliated with the funding organization, indicating the funder may have had some role in the design, data collection, analysis or preparation of your manuscript for publication; in other words, the funder played an indirect role through the participation of the co-authors. If the funding organization did not play a role in the study design, data collection and analysis, decision to publish, or preparation of the manuscript and only provided financial support in the form of authors' salaries and/or research materials, please do the following:

(1) Review your statements relating to the author contributions, and ensure you have specifically and accurately indicated the role(s) that these authors had in your study. These amendments should be made in the online form.

(2) Confirm in your cover letter that you agree with the following statement, and we will change the online submission form on your behalf: 

"I have read the journal's policy and the authors of this manuscript have the following competing interests: Associate Professor Moore is in receipt of research funds from Merck Sharp and Dohme (MSD) and Sanofi (unrelated to the work presented in this paper). Associate Professor Moore has also received institutional honoraria for participating in advisory committees (Pfizer, MSD, Sanofi), also unrelated to the work presented in this paper. Associate Professor Gidding has received honoraria for participating in a Seqirus advisory committee unrelated to the work presented in this paper. Professor Richmond has served on pertussis vaccine scientific advisory boards for GlaxoSmithKline and Sanofi on behalf of his institution. He also participated in multicenter vaccine trials of pertussis vaccines sponsored by industry. He has received no personal remuneration for these activities. No other disclosures were reported."

We note that you received funding from a commercial source: Merck Sharp and Dohme (MSD), Sanofi, Pfizer and GlaxoSmithKline.

Within this Competing Interests Statement, please confirm that this does not alter your adherence to all PLOS ONE policies on sharing data and materials by including the following statement: "This does not alter our adherence to PLOS ONE policies on sharing data and materials.” (as detailed online in our guide for authors http://journals.plos.org/plosone/s/competing-interests).  

If there are restrictions on sharing of data and/or materials, please state these. Please note that we cannot proceed with consideration of your article until this information has been declared. 

5. Please upload a new copy of Figure 1 and Supporting Figure S1 as the detail is not clear. Please follow the link for more information: https://blogs.plos.org/plos/2019/06/looking-good-tips-for-creating-your-plos-figures-graphics/

https://blogs.plos.org/plos/2019/06/looking-good-tips-for-creating-your-plos-figures-graphics/

7. In your Data Availability statement, you have not specified where the minimal data set underlying the results described in your manuscript can be found. PLOS defines a study's minimal data set as the underlying data used to reach the conclusions drawn in the manuscript and any additional data required to replicate the reported study findings in their entirety. All PLOS journals require that the minimal data set be made fully available. For more information about our data policy, please see http://journals.plos.org/plosone/s/data-availability.

**Additional Editor Comments:**

Thank you for submitting your manuscript. The reviewers and I believe it is of potential value for our readers. However, the reviewers have raised a number of very important issues, and their excellent comments will need to be adequately addressed in a revision before the acceptability of your manuscript for publication in the Journal can be determined. We cannot guarantee that your revised paper will be chosen for publication; this would be solely based on how satisfactorily you have addressed the reviewer comments.

Reviewers' comments:

Reviewer's Responses to Questions

**Comments to the Author**

1. Does the manuscript adhere to the experimental procedures and analyses described in the Registered Report Protocol?

If the manuscript reports any deviations from the planned experimental procedures and analyses, those must be reasonable and adequately justified.

Reviewer #1: Partly

Reviewer #2: Yes

2. If the manuscript reports exploratory analyses or experimental procedures not outlined in the original Registered Report Protocol, are these reasonable, justified and methodologically sound?

A Registered Report may include valid exploratory analyses not previously outlined in the Registered Report Protocol, as long as they are described as such.

Reviewer #1: No

Reviewer #2: Yes

3. Are the conclusions supported by the data and do they address the research question presented in the Registered Report Protocol?

The manuscript must describe a technically sound piece of scientific research with data that supports the conclusions. The conclusions must be drawn appropriately based on the research question(s) outlined in the Registered Report Protocol and on the data presented.

Reviewer #1: Yes

Reviewer #2: Yes

4. Have the authors made all data underlying the findings in their manuscript fully available?

Reviewer #1: No

Reviewer #2: Yes

5. Is the manuscript presented in an intelligible fashion and written in standard English?

Reviewer #1: Yes

Reviewer #2: Yes

6. Review Comments to the Author

Please use the space provided to explain your answers to the questions above. (Please upload your review as an attachment if it exceeds 20,000 characters)

Reviewer #1: Reviewer response

Association between pertussis vaccination in infancy and asthma: a population-based record linkage cohort study

comments

• I really appreciate the research idea however the topic needs a bit of improvement.

Introduction section

It is well written however it is more clinical hence authors need to show the asthma burden among school children, what are the health consequences, and the economic burden of it.

Why did you prefer to conduct ages between 5 to 15 years children?

Your outcome variable is not clearly stated so you need to write clearly again.

I didn’t see the clearly stated inclusion and exclusion criteria so you better follow the guideline of the journal.

I found two outcome variables such as time to 1st event and time to recurrent event so how did you manage it? I think it is somewhat vast. Both might have different factors that you have to analyze, show results, and discuss separately.

There is no independent operational definition section and you fail to operationalize some concepts that you mention in the document for instance residual confounder, first event of time, recurrent event

Regarding method section

I didn’t find a data extraction tool with its description. I think it has to be incorporated in this section.

Again, you fail to describe how to control data quality so you have to write it.

In the data analysis and process section: some indispensable points are overlooked or not written like model fitness, what were you assumption to take candidate variables from bivariate to multivariate Cox regression statistical model, VIF, AND proportional hazard assumptions?

Result section

How much data you enrolled from that how much data you analyzed?

There are two tables that existed in your document but it doesn’t show the succession which means you put table 1 and table 1 so at least you need to show the continuity of the table by using the word “table 1 continued”.

Since your study is about survival, where are the important graphs that show your study like Kaplan Meir survival estimate, cox regression table, and Comparison of survivorship functions for different categorical variables?

Discussion section

It is shallow to reveal the detail of your findings

References

Of the total of 3,4,13,14,15,20, and 22 references are outdated which is not appropriate.

Reviewer #2: Dear authors,

I have now completed the review of the manuscript titled "Association between pertussis vaccination in infancy and asthma: a population-based

record linkage cohort study."

In the present study, the authors evaluated hypothesis that whole-cell pertussis (wP) vaccination in early infancy might protect against atopic asthma in childhood.

The manuscript is interesting and, in general, fair written.

I have some suggestions to further improve the quality of the manuscript.

1. The background section introduced some relevant articles. Please explain the results or summarize with effect sizes.

2. Authors used ICD-10 AM diagnostic codes. I wonder do they also have old ICD-9 codings or if all of the data were originally set to ICD-10.

3. In the ‘Statistical Analysis’ section, I suggest authors to also refer open access articles than current closed access articles, for readers who are interested.

4. What is the future scope of the proposed research, authors have described the limitations in a good way, and I suggest that these can be the future scope of the work.

7. PLOS authors have the option to publish the peer review history of their article (what does this mean?). If published, this will include your full peer review and any attached files.

Reviewer #1: No

Reviewer #2: No

---

## [Author Response · Author response to Decision Letter 0]

9 Aug 2023

Our response to the reviewers has been uploaded as an attachment as it exceeds 20,000 characters

---

## [Decision Letter · Decision Letter 1]

22 Aug 2023

PONE-D-23-11079R1Association between pertussis vaccination in infancy and  childhood asthma: a population-based record linkage cohort studyPLOS ONE

Dear Dr. Pérez Chacón,

Thank you for submitting your manuscript to PLOS ONE. After careful consideration, we feel that it has merit but does not fully meet PLOS ONE’s publication criteria as it currently stands. Therefore, we invite you to submit a revised version of the manuscript that addresses the points raised during the review process.

We look forward to receiving your revised manuscript.

Kind regards,

Dong Keon Yon, MD, FACAAI, FAAAAI

Academic Editor

PLOS ONE

Journal Requirements:

Additional Editor Comments:

This is an excellent paper. Please see my minor comments.

#1. Primary and secondary complete-case analyses (i.e., including only observations with no missing data) were performed to estimate unadjusted and adjusted hazard ratios (HRs), and their 95% CIs -> Please cite the statistical guideline (DOI: https://doi.org/10.54724/lc.2023.e8).

#2. Please add this sentence. A two-sided P less than 0.05 considered significant.

#3. Please add IRB registration number.

This study was approved by the human research ethics committees of the Department of Health of WA, NSW Population Health Service, Australian Institute of Health and Welfare, Curtin University (Number XXX), the WA Aboriginal Health Ethics Committee (Number XXX), and the Aboriginal Health and Medical Research Council Ethics Committee (Number XXX).

This is an excellent paper!!

Reviewers' comments:

Reviewer's Responses to Questions

**Comments to the Author**

1. Does the manuscript adhere to the experimental procedures and analyses described in the Registered Report Protocol?

If the manuscript reports any deviations from the planned experimental procedures and analyses, those must be reasonable and adequately justified.

Reviewer #2: Yes

2. If the manuscript reports exploratory analyses or experimental procedures not outlined in the original Registered Report Protocol, are these reasonable, justified and methodologically sound?

A Registered Report may include valid exploratory analyses not previously outlined in the Registered Report Protocol, as long as they are described as such.

Reviewer #2: Yes

3. Are the conclusions supported by the data and do they address the research question presented in the Registered Report Protocol?

The manuscript must describe a technically sound piece of scientific research with data that supports the conclusions. The conclusions must be drawn appropriately based on the research question(s) outlined in the Registered Report Protocol and on the data presented.

Reviewer #2: Yes

4. Have the authors made all data underlying the findings in their manuscript fully available?

Reviewer #2: Yes

5. Is the manuscript presented in an intelligible fashion and written in standard English?

Reviewer #2: No

6. Review Comments to the Author

Please use the space provided to explain your answers to the questions above. (Please upload your review as an attachment if it exceeds 20,000 characters)

Reviewer #2: All comments were addressed. Thank you for authors and editors considering my opinion on this manuscript.

7. PLOS authors have the option to publish the peer review history of their article (what does this mean?). If published, this will include your full peer review and any attached files.

Reviewer #2: No

---

## [Author Response · Author response to Decision Letter 1]

26 Aug 2023

Response to the Editor’s comments

This is an excellent paper. Please see my minor comments.

• Response: We thank the Editor for their feedback and comments.

#1. Primary and secondary complete-case analyses (i.e., including only observations with no missing data) were performed to estimate unadjusted and adjusted hazard ratios (HRs), and their 95% CIs -> Please cite the statistical guideline (DOI: https://doi.org/10.54724/lc.2023.e8).

• Response: The revised version of our manuscript includes the above-mentioned citation (i.e., ref 24).

#2. Please add this sentence. A two-sided P less than 0.05 is considered significant.

• Response: We value the suggestion from the Editor. Nevertheless, given the multiplicity of testing, we are refraining from designating a p-value of <0.05 as indicating statistical significance.

#3. Please add IRB registration number. This study was approved by the human research ethics committees of the Department of Health of WA, NSW Population Health Service, Australian Institute of Health and Welfare, Curtin University (Number XXX), the WA Aboriginal Health Ethics Committee (Number XXX), and the Aboriginal Health and Medical Research Council Ethics Committee (Number XXX). This is an excellent paper!!

• Response: We thank the Editor for their feedback. We have edited this section and included the approval numbers: This study was approved by the human research ethics committees of the Department of Health of WA, (approval number: 2012/75), NSW Population Health Service, (approval number: HREC/13/CIPHS/15), Australian Institute of Health and Welfare, (approval number: EC2012/4/62), Curtin University, (approval number: HRE2019-0350), the WA Aboriginal Health Ethics Committee, (approval number: 459), and the Aboriginal Health and Medical Research Council Ethics Committee (approval number: 931/13). A waiver of consent was requested and granted owing to the large size of the study cohort.

---

## [Editor Report · Decision Letter 2]

31 Aug 2023

Association between pertussis vaccination in infancy and  childhood asthma: a population-based record linkage cohort study

PONE-D-23-11079R2

Dear Dr. Pérez Chacón,

We’re pleased to inform you that your manuscript has been judged scientifically suitable for publication and will be formally accepted for publication once it meets all outstanding technical requirements.

Kind regards,

Dong Keon Yon, MD, FACAAI, FAAAAI

Academic Editor

PLOS ONE

Additional Editor Comments (optional):

This is an excellent paper.
---

## [Editor Report · Acceptance letter]

26 Sep 2023

PONE-D-23-11079R2 

Association between pertussis vaccination in infancy and childhood asthma: a population-based record linkage cohort study 

Dear Dr. Pérez Chacón:

I'm pleased to inform you that your manuscript has been deemed suitable for publication in PLOS ONE. Congratulations! Your manuscript is now with our production department. 

Kind regards, 

on behalf of

Dr. Dong Keon Yon 

Academic Editor

PLOS ONE